# Experimental and DFT Study of Monensinate and Salinomycinate Complexes Containing {Fe_3_(µ_3_–O)}^7+^ Core

**DOI:** 10.3390/molecules29020364

**Published:** 2024-01-11

**Authors:** Nikolay Petkov, Alia Tadjer, Elzhana Encheva, Zara Cherkezova-Zheleva, Daniela Paneva, Radostina Stoyanova, Rositsa Kukeva, Petar Dorkov, Ivayla Pantcheva

**Affiliations:** 1Faculty of Chemistry and Pharmacy, Sofia University St. Kliment Ohridski, 1164 Sofia, Bulgaria; tadjer@chem.uni-sofia.bg (A.T.); eencheva@ipc.bas.bg (E.E.); 2Institute of Physical Chemistry, Bulgarian Academy of Sciences, 1113 Sofia, Bulgaria; 3Institute of Catalysis, Bulgarian Academy of Sciences, 1113 Sofia, Bulgaria; zzhel@ic.bas.bg (Z.C.-Z.); daniela@ic.bas.bg (D.P.); 4Institute of General and Inorganic Chemistry, Bulgarian Academy of Sciences, 1113 Sofia, Bulgaria; radstoy@svr.igic.bas.bg (R.S.); rositsakukeva@yahoo.com (R.K.); 5Research and Development Department, Biovet Ltd., 4550 Peshtera, Bulgaria; p_dorkov@biovet.com

**Keywords:** polyether ionophorous antibiotics, iron(III) oxo-complexes, EPR, Mössbauer, molecular modelling

## Abstract

Two trinuclear oxo-centred iron(III) coordination compounds of monensic and salinomycinic acids (HL) were synthesized and their spectral properties were studied using physicochemical/thermal methods (FT–IR, TG–DTA, TG–MS, EPR, Mössbauer spectroscopy, powder XRD) and elemental analysis. The data suggested the formation of [Fe_3_(µ_3_–O)L_3_(OH)_4_] and the probable complex structures were modelled using the DFT method. The computed spectral parameters of the optimized constructs were compared to the experimentally measured ones. In each complex, three metal centres were joined together at the axial position by a μ_3_–O unit to form a {Fe_3_O}^7+^ core. The antibiotics monoanions served as bidentate ligands through the carboxylate and hydroxyl groups located at the termini. The carboxylate moieties played a dual role bridging each two metal centres. Hydroxide anions secured the overall neutral character of the coordination species. Mössbauer spectra displayed asymmetric quadrupole doublets that were consistent with the existence of two types of high-spin iron(III) sites with different environments—two Fe[O_5_] and one Fe[O_6_] centres. The solid-state EPR studies confirmed the +3 oxidation state of iron with a total spin S_t_ = 5/2 per trinuclear cluster. The studied complexes are the first iron(III) coordination compounds of monensin and salinomycin reported so far.

## 1. Introduction

Metal ions are essential for maintaining homeostasis in organisms ranging from the microscopic world of bacteria to the macroscopic scale of plants, animals, and, of course, human beings [1,2,3,4]. The importance of M^n+^ is rooted not in their varying concentration in each organism, but in the pathologies arising from an abnormal amount of certain metal ions in a given life form. Special attention must be paid to microelements, as variations in their concentration can cause severe health issues.

Iron ions play a vital role in almost all living organisms [5] (except *Lactobacillus* and some strains of *Bacillus* [6]). Among the special functions of iron are the transport, storage and activation of molecular oxygen, reduction in ribonucleotides and dinitrogen, activation and decomposition of peroxides, and electron transport using various electron carriers [7]. On the other hand, iron biochemistry can be crucial for cancer occurrence and development since an abnormal regulation in cancer cells has been observed—downregulated efflux of iron ions from the cells, generation of reactive oxygen species (ROS), ferroptosis induction, etc. [8,9,10,11,12,13,14,15,16,17,18,19].

If we look at some of the simplest living organisms, for example, representatives of microbiota, we witness an interesting natural way of scavenging insoluble iron(III) from the environment—these organisms synthesize specific compounds called siderophores [1,20,21,22,23,24,25,26,27], which effectively bind iron(III) and transfer it to the intracellular compartment. Based on their structures, siderophores have been divided into five groups—catecholates, phenolates, hydroxamates, carboxylates, and a mixed type [27]. Their primary role is to ensure iron(III) uptake even at low concentration levels. This process also occurs in infected humans—the so-called “nutritional immunity” triggers and the host organism itself limits the amount of trace elements accessible to the pathogen [9]. The result is a dramatic change in the metabolism of the invader and its subsequent cell death.

Compounds that bind iron and transport it through the cells, without ion release or changing its reduction potential, can function as chemotherapeutics. Deferoxamine and deferasirox are two such compounds that are known to exclusively bind iron ions [11,16,28]. These drugs have already been included in therapeutic protocols and are used in the case of iron overload in some haematological diseases, metal intoxication, and in the treatment of aceruloplasminemia patients. The availability of such approved compounds in human medicine raises the question of whether a synergistic effect could be achieved by using a metal chelator as a ligand which can bind iron ions, but also can specifically induce apoptosis in cancer cells or/and cancer stem cells as well [10,11,13,14,15,16,19,29,30].

Such biologically active compounds can be found among the ionophore antibiotics, which have been in the spotlight for the last decade due to their anticancer properties. A member of this group—salinomycin, shows a substantial effect on cancer cells and cancer stem cells [18,31,32]. Another representative is monensin, which possesses less pronounced anticancer activity compared to salinomycin but has potential to be included in cancer therapy, due to the high selectivity index of some of its recently studied metal complexes [33]. Both antibiotics are polyether derivatives of monocarboxylic acids and, similarly to carboxylate siderophores, they may serve as potential ligands for iron(III) binding.

In this study, the ability of the natural antibiotics monensin (MonH) and salinomycin (SalH) (Figure 1) to interact with iron(III) was studied via a plethora of physicochemical methods. Two isostructural coordination compounds bearing the corresponding ionophore were synthesized. The resulting complexes consisted of two types of ligands directly bound to metal ions situated in a trinuclear configuration typical for iron(III) monocarboxylates. The composition and the binding mode of the ligands were proposed based on experimental data. Essential information for the primary coordination shell and the electronic configuration of the newly synthesized complexes was derived from EPR and Mössbauer spectroscopy. The mutual position of the atoms in the cores of the complexes was evaluated by appropriate theoretical models. Due to the complicated structure of polyether ionophores, solely the closest environment around the metal centres was modelled. The computed data obtained for the simplified system revealed the spatial placement of the metal cations and the donor atoms but also disclosed the magnetic properties of the complexes, which largely impact their spin state population at room temperature. The theoretically predicted electronic structure and properties of iron(III) monensinate and salinomycinate were used to explain the experimentally observed results and some peculiarities in the spectral behaviour of the coordination compounds.

## 2. Results and Discussion

### 2.1. General

Iron(III) monensinate (**1**) and salinomycinate (**2**) were obtained from methanolic solutions containing MonH×H_2_O or SalH, respectively, Et_3_N and FeCl_3_×6H_2_O. Precipitation from water produces yellow coordination species in satisfactory yields. The good agreement between the experimental and theoretical data allowed us to deduce plausible structures of novel iron(III) complexes of monensin (**1**) and salinomycin (**2**) as trinuclear oxo-clusters of composition [Fe_3_(µ_3_–O)L_3_(OH)_4_].

### 2.2. Elemental and Thermal Analysis

The elemental analysis data of the isolated complexes **1**–**2** can be found in Table 1. The thermal studies (TG–DTA and TG–MS) exclude the presence of coordinated water molecules in the composition of the solid samples (Appendix A). The powder XRD analysis (Appendix A) reveals the lack of crystallinity in both complexes evident from their amorphous halo located at small angles. The experimental data indicate the formation of isostructural equimolar coordination species of both antibiotics (Fe^3+^:ligand ratio 1:1), the absence of water, and the direct involvement of hydroxide anions in the close environment of the metal ions.

### 2.3. IR Spectroscopy

The free non-coordinated acidic forms of the polyether ionophores possess characteristic bands in the IR region, assigned to stretching vibrations of the hydroxyl (4000–3000 cm^−1^), carboxyl (1710 cm^−1^), and carbonyl (1710 cm^−1^, SalH) groups.

The band at 1710 cm^−1^ disappears (monensinate, Mon^−^) or significantly diminishes in intensity (salinomycinate, Sal^−^) when switching to the spectra of **1**–**2**. Instead, two new bands in the range of 1530–1520 cm^−1^ and 1420 cm^−1^ are observed attributed to ν_as_(−COO^−^) and ν_s_(−COO^−^), respectively (Figure 2). The difference between the two vibrations is 112 (**1**) and 103 (**2**) cm^−1^ and is consistent with the presence of a bridging deprotonated carboxyl function in the ionophores structure [34,35,36]. The strong absorption peaks at ca. 1050 cm^−1^ are characteristic for δ(FeOH), suggesting the inclusion of hydroxide anions as additional ligands in the primary coordination shell of the metal centres. The peaks at ca. 435 cm^−1^ in the spectra of **1**–**2** are specific for ν(FeO), pointing to the formation of Fe–O bonds [37]. The broad intense bands in the IR spectra of **1**–**2** centred at 3470–3465 cm^−1^ are assigned to various ν(OH) vibrations.

### 2.4. EPR Spectroscopy

The EPR spectra of **1**–**2** presented in Figure 3a,b were recorded within the temperature range 77–295 K. Both spectra consist of one relatively broad symmetric signal with Lorentz shape. Its peak-to-peak linewidths at room temperature are 99 mT (**1**) and 122 mT (**2**) and undergo identical broadening as the temperature decreases (Figure 3c,d). The signal position is temperature independent, and the g-factor values remain in a very narrow interval in the whole temperature range (2.020–2.005 for **1** and 2.030–2.010 for **2**). Further study of the temperature dependence of the signal intensity reveals that the Currie–Weiss law is obeyed in the studied samples and that the experimentally estimated values of the Currie–Weiss constant are as follows: −378 ± 12 K (**1**); −380 ± 9 K (**2**). The negative Currie–Weiss constants and the signal broadening at low temperatures indicate an antiferromagnetic interaction between the magnetically coupled Fe^3+^ ions. The linewidth temperature dependence could be explained by increased magnetic interactions at low temperatures. The detection of a broad EPR signal for complexes **1** and **2**, with a g-factor around 2.0, is due to exchange-coupled individual Fe^3+^ ions (^6^S_5/2_) [38]. In most ferric monocarboxylate complexes these interactions occur through an oxo-anion centred in the core of the coordination compounds. Since the EPR characteristics of our complexes are consistent with the concept of a trinuclear antiferromagnetically coupled cluster of Fe^3+^ ions and based on the IR and elemental analysis data, the species **1**–**2** can be described as oxo-complexes of the antibiotics, namely [Fe_3_(µ_3_-O)L_3_(OH)_4_].

### 2.5. Mössbauer Spectroscopy

The zero-field Mössbauer spectra of both samples at 293 K consist of asymmetric doublets that can be decomposed into two Lorentzian-shaped quadrupole doublets with an area ratio of 2:1 (Table 2, Figure 4). A least-square fit of the spectra yields isomer shifts (IS, δ) within the range of 0.3 to 0.6 mm/s typically observed for high-spin iron(III) complexes [39,40]. The half-filled 3d electron level implies that the charge distribution asymmetry around the metal centres dominantly contributes to the observed quadrupole splitting (QS, ∆). The QS of the two doublets, which describes the experimental signals of **1**–**2**, leads to the conclusion that two of the ions possess more symmetric charge distribution (ca. 0.60) than the third one (ca. 0.80). The temperature decrease does not change significantly the QS values (Table 2, Appendix A), but leads to an increase in the IS values, attributable to a second-order Doppler effect. The noticeably more prominent Mössbauer effect can be explained in terms of decreased vibrational motions of the molecules, thus increasing the probability of recoilless absorption of the gamma quant.

The subspectra with an intensity ratio of 2:1 (Figure 4) can be assigned to two penta-coordinated ions Fe(1) and Fe(2) (subspectrum (1)) and one six-coordinated ion Fe(3) (subspectrum (2)), respectively, as it is shown in the sketch, presented in Figure 5. The isomer shifts, together with the quadrupole splittings, support the high-spin state for the three iron sites. The similarity in the isomer shifts of Fe(l)/Fe(2) and Fe(3) might be assigned to close bond lengths at all irons. The penta-coordinated ferric ions Fe(1) and Fe(2) are indistinguishable in the Mössbauer spectra while the quadrupole splitting of the six-coordinated Fe(3) exceeds the value of Fe(1) and Fe(2) [41].

In summary, we were able to isolate two new Fe(III) complexes of monensin (**1**) and salinomycin (**2**). The spectral data reveal the following:(i)The antibiotics undergo deprotonation and act as bridging monoanions (IR);(ii)No water ligands are included in the primary coordination shell of the metal ions (TGV);(iii)An antiferromagnetic interaction between the metal centres occurs that can be realized within a triangular oxo-iron cluster (EPR);(iv)The three high-spin iron ions are inequivalent and possess non-symmetric charge distribution at 2:1 ratio (Mössbauer spectroscopy).

Based on the experimental evidence, we assume that the new coordination species belong to the known class of trinuclear oxo-ferri-carboxylates with [Fe_3_(O)L_3_]^4+^ composition. Most likely, each ionophore serves as a bridge between every two metal centres through the carboxylate function, but also acts as a bidentate ligand via the hydroxyl group placed at the opposite end of the molecule. The overall neutral character of the coordination species formed is ensured by hydroxide anions originating from the basic medium used for the complexes preparation. Three of these anions occupy the hydrophilic cavity of antibiotics and bind to irons completing their penta-coordinated shell. In addition, an extra hydroxide is bound to one of the metal centres to fulfil its sixth coordination position (Fe(3) in Figure 5).

Such “head-to-tail” cyclization accompanied by a cave-hosted water molecule or hydroxide ion is a common feature of polyether ionophores [33,42,43], but the unambiguous coordination mode and the folding of the antibiotics can only be proved by X-ray diffraction on monocrystals. Unfortunately, our attempts to grow suitable crystals were unsuccessful; that is why we used several spectral techniques to explore the structure of the target complexes. The magnetic data disclose the antiferromagnetic properties of **1**–**2**, which can arise only through an indirect coupling, namely via an oxo-anion centred into the core of a trinuclear iron cluster. For deeper insight into the structure of the new iron(III) monensinate and salinomycinate, we carried out theoretical studies to rationalize some of the experimentally observed peculiarities.

### 2.6. Theoretical Studies

The entire structures of complexes **1**–**2** are oversized for reliable first principles quantum chemical calculations. Therefore, we limited ourselves to modelling the primary coordination shell of the iron ions. The crystal structure of ferric acetate [44] was used as a template and three of the acetate ligands were replaced by ethanol molecules to reproduce the antibiotics “tail” hydroxyl groups. The experimental evidence that our complexes contain no coordinated water and that the three irons are placed in 2:1 different environments prompted us to remove two water ligands in the initial structure and to change the last one to a hydroxide anion. In addition, we built two possible architectures that differ in the orientation of the carboxylate anion: in the first one the COO^−^-groups are parallel to the plane of the {Fe_3_(μ_3_–O)} core (Figure 6A), while in the second one the COO^−^-function bridging Fe(1) and Fe(2) lies outside this plane (Figure 6B). The stability of the two structures was assessed by the full geometry optimization of the corresponding high spin states (HS, S_t_ = 15/2). The free energy difference between **A** and **B** is ca. 6 kcal/mol which indicates that construct **B** is the preferred configuration. Moreover, **B** is the more reasonable one, if we consider the antibiotics total skeletons and the geometry of their known compounds [43]. Structural characteristics of **B** are shown in Table 3. The iron ions Fe(1) and Fe(2) are situated in triangular bipyramidal environments with Fe−O bond lengths varying within the range of 1.9–2.1 Å, typical for iron oxygen-containing compounds [44,45,46,47,48]. The third metal centre Fe(3) is enveloped by six oxygens which form a slightly distorted octahedral structure, where the equatorial plane is occupied by equidistant carboxylate, alcohol, and hydroxide oxygens, while the second hydroxide and the oxo-oxygens are placed at axial positions. The inequivalent surrounding of the individual metal centres corroborates well the experimentally observed Mössbauer asymmetric doublets for **1**–**2**.

Fe(III) features five unpaired electrons; this allows different multiplicities and magnetic behaviour of the trinuclear construct—ferromagnetic, with total spin S_t_ = 15/2 (labelled HS) and antiferromagnetic with total spin S_t_ = 5/2 (labelled AFMSx). To model the possible antiferromagnetic sextets, we applied a spin flip on each metal centre (the corresponding AFM sextets labelled according to the Fe numbering in Figure 5 and Figure 6). This procedure does not change the geometry but changes the energetics and the electron and spin distribution. The Gibbs free energy at 293 K was calculated and each state population in the range 0−450 K (Figure 7) was determined (450 K being approximately the melting temperature of the complexes). AFMSx_1 and AFMSx_2 exhibit ignorable energy difference of less than 0.1 kcal/mol and are essentially degenerate and structurally identical; so only structural results for AFMSx_1 are shown further on. Thus, the most unfavourable state is the high-spin one (S_t_ = 15/2) followed by AFMSx_3. The spin population is temperature dependent: at room temperature all species are present while at 77 K and 0 K only the penta-coordinated irons define the magnetic conduct. Yet, the calculated g-factors are very close (Table 4). These findings are in agreement with the EPR data which imply an enhanced antiferromagnetic interaction between magnetically coupled iron(III) ions upon temperature decrease. In addition, the credibility of the presence of construct **B** as antiferromagnetic sextet is supported by the calculated g-factor of 2.005, which matches the experimentally observed ones g = 2.005 (**1**) and g = 2.010 (**2**), respectively.

The NBO spin population of the [Fe_3_(μ_3_–O)]^7+^ core is presented in Table 5. It is easily seen that the spin distribution of the iron ions is analogous in all spin states—slightly lower at the penta-coordinated state and always somewhat higher at the octahedral species. The missing portion of the total spin is distributed predominantly on the oxygens with highest values (0.1−0.2) on the equatorial hydroxide one. The spin density on the (μ_3_–O)-centre correlates with the relative energy of the spin states (Table 4) indicating that the complex stability depends critically on the spin localization at this centre.

Similar to the spin distribution, the charge distribution in the first coordination shell (Table 6) is fairly uniform: all iron ions bear charges of ca. 2.2 (minor excess on the six-coordinated one), all carboxylate oxygen charges are ca. −0.8, the alcohol groups are with summed charges −0.3, and the hydroxide ions are with total charge −0.7. This confirms the overall identical environment of Fe(1) and Fe(2). The charge envelope of Fe(3) differs only in the additional hydroxide ligand. Unlike the spin density distribution, the charge on the μ_3_–O fragment is constant in all spin states.

The s-electron density on the iron ions, calculated from the NAO occupancy, is 6.246, irrespective of the coordination pattern, and corroborates the identical isomer shift established from the Mössbauer spectra analysis (Table 2, Figure 4).

Valuable information regarding the solidity of the bonds in the first coordination sphere of the complexes can be obtained from the vibrational analysis of the most stable structure. To be able to compare the computed to the experimentally measured frequencies for **1**–**2**, we need to make models that differ in mass, as do the real ligands, while conserving the simplicity of the constructs. For this purpose, isotopic substitution of some carbons and hydrogens in structure **B** was performed. When choosing the isotopes and their arrangement, a maximally symmetric distribution was maintained with respect to the main chain of the model ligand, as well as the mass number of the isotope used. ^2^H, ^13^C, and ^14^C were chosen (Appendix A), thereby minimizing direct impact on the atoms directly involved in the metal–oxygen bonds.

The most important vibrational frequencies are presented unscaled in Table 7 and compared to the recorded IR spectra of **1**–**2**. The deviations are within an acceptable range, and the trends witnessed in the experimental data (Figure 2) are preserved. Lower values for the vibrational frequencies in salinomycin directly correlate with the larger effective mass of the ligand. This allows us to see a correlation between the Mössbauer effect observed and the bond strength on the other hand. The vibrational frequencies of the carboxylate groups are in excellent agreement with the experimentally recorded ones and with the differences in the wave numbers for asymmetric and symmetric stretching, which confirms the validity of the model and the assumption about the structure of the first coordination sphere. Moreover, this correspondence authenticates the isotopic substitution as an approach for distinctive treatment, with respect to predicting the vibrational behaviour of large systems, having an identical core but different mass, which, to our knowledge, has not been reported so far.

## 3. Materials and Methods

### 3.1. Reagents

Sodium monensinate (MonNa) and salinomycinate (SalNa) were complimentary supplied by Huvefarma Ltd. (Peshtera, Bulgaria). Monensic acid (MonH×H_2_O) and salinomycinic acid (SalH) were prepared according to Gertenbah and Popov [49]. The rest of the reagents of p.a. grade (FeCl_3_×6H_2_O, Et_3_N, acetonitrile, methanol, and ether) were purchased from local suppliers. In all experiments deionized water was used.

### 3.2. Synthesis of Complexes **1**–**2**

MonH×H_2_O (0.5 mmol, 344 mg in 15 mL MeOH) or SalH (0.5 mmol, 375.5 mg), respectively, was mixed with Et_3_N (1 mmol, 139.5 µL) and stirred at r. t. for 15 min. After that FeCl_3_×6H_2_O (0.25 mmol, 70.1 mg in 15 mL MeOH) was slowly added and the resulting orange solution was precipitated from water to produce a yellow solid phase, which was filtered off, washed subsequently with water and ether, and dried in a desiccator. The composition and elemental analysis data for coordination compounds **1** (MonH) and **2** (SalH) are presented in Table 1. Yield: **1**—281 mg, 75%; **2**—292 mg, 70%. The isolated complexes are insoluble in water and are sparingly soluble in most of the organic solvents.

### 3.3. Spectroscopic Studies

Infrared (IR) spectra were obtained in KBr pellets (4000–400 cm^−1^) using a Nicolet 6700 FT-IT spectrometer (Thermo Scientific, Madison, WI, USA).

Thermogravimetric experiments (TG−DTA/MS) were performed on Setaram Labsys Evo 1600 (25–600 °C) with a heating rate of 10 K/min in argon flow (Caluire-et-Cuire, France). The apparatus is equipped with an Omnistar GSD 301 O2 mass spectrometer, Pfeiffer Vacuum.

^57^Fe Mössbauer measurements were performed using a Wissel spectrometer (Wissenschaftliche Elektronik GmbH, Starnberg, Germany), operating in a constant acceleration mode. The transmission Mössbauer spectra were measured at room temperature (293 K) and 77 K with an analyser with 1024 channels, source ^57^Co/Rh (15 mCi). The samples were adjusted to an optimum thickness for a “thin” absorber density of about 10 mg/cm^2^ of the total iron content (60 mg of solid samples studied). The hyperfine interaction parameters: isomer shift (δ) relative to α-Fe and quadrupole splitting (∆) were determined via computer fitting with Lorentzian lineshapes using the program NORMOS (based on the least squares method). The isomer shift (δ), quadrupole splitting (∆), and line width (Γ) are calculated with ±0.01 mm/s error.

The electron paramagnetic resonance (EPR) studies were conducted on Bruker BioSpin EMXplus10/12 EPR spectrometer (Karlsruhe, Germany) working at 9.4 GHz.

The sample crystallinity degree was evaluated by powder X-ray diffraction (XRD) analysis using PANalytical Empyrean X-ray diffractometer (Malvern Panalytical, Malvern, UK) with CuKα radiation (λ = 0.15418 nm) operating at 40 kV, 30 mA.

Elemental analysis (C, H) was performed on an organic elemental analyser Vario MACRO cube (Elementar analysensysteme GmbH, Stuttgart, Germany). The metal content was determined on Perkin-Elmer SCIEX-ELAN DRC-e ICP-MS (Waltham, MA, USA) after wet digestion of samples with conc. HNO_3_ using appropriate standards.

### 3.4. Quantum Chemical Calculations

The overall structure of the complexes **1**–**2** requires a large number of basis functions for its description, which exceeds the potential of the computational software and hardware available. For this reason, simplifications were considered and only the first coordination sphere was modelled as detailed in Section 2.6. The geometry of all structures in states with multiplicities 6 (AFMSx) and 16 (HS) was optimized with the hybrid functional B3LYP [50], employing the 6-31G(d) basis set, in vacuo, including the Grimme D3 correction for dispersion interactions within the Gaussian 16 software. The energy minimum of each structure was verified with a vibrational frequency analysis. These structures were a starting point for the calculation of all energies and EPR properties using the ORCA 5.0.3 software package. Based on the geometries obtained and their symmetry, a further spin-flip on each iron centre was also performed. The calculations were meant to provide reliable results for high-spin systems in which the exchange interactions are of paramount importance, which is why we switched to a hybrid functional with 50% Hartree–Fock exchange such as BHandHLYP [51,52,53] with a triple-zeta basis set for the ligands and CP(PPP) basis set for Fe [54]. All calculations were performed in vacuo. For each structure and multiplicity, the thermochemical properties and EPR parameters were evaluated. Particularly, the free energy comprises contributions of the total energy, the zero-point vibrational energy, the translational, rotational, and vibrational energy to the entropy assessment at the respective temperature, the latter energy quantified in the harmonic approximation.

## 4. Conclusions

Aiming to achieve the combined effect of iron binding agents and cancer therapeutics, the first complexes of Fe(III) with two antibiotics with promising anticancer activity were synthesized and characterized structurally utilizing an assortment of spectroscopies and first principles molecular modelling. The antibiotics monensin and salinomycin have kindred molecular skeletons and documented “head-to-tail” patterns of coordination with d-elements. The complexes obtained in alkaline medium do not form crystals for X-ray description; therefore, the vibrational, Mössbauer, and EPR spectroscopies were employed for characterization. The elemental analysis revealed 1:1 ratio of Fe ions and ionophorous antibiotics. The hypothesis that the structure of the complexes contains a trinuclear {Fe_3_(µ_3_–O)}^7+^ core allowed their simulating with a simple molecular model, for which quantum chemical methods of trustworthy quality could be applied. The agreement between the theoretically assessed and the experimentally measured characteristics is very good, validating the coordination pattern and the antiferromagnetic spin state. In addition to the structural features, molecular modelling shed light on the charge and spin distribution and on the temperature dependence of the spin states population, reflected in the physicochemical characteristics variation upon temperature decrease. Furthermore, it was shown that isotopic substitution is an efficient strategy for predicting the vibrational behaviour of compounds with the same core but with ligands of different mass; this can be deemed a methodological merit of the modelling approach in the study.

## Figures and Tables

**Figure 1 molecules-29-00364-f001:**
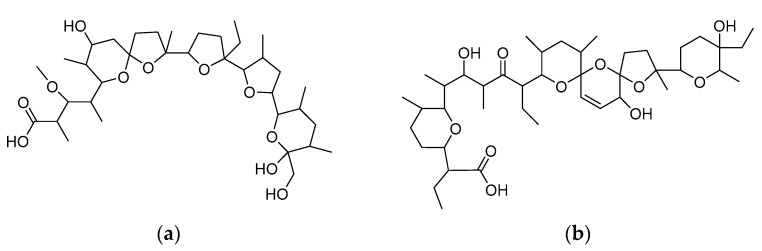
Chemical structures of (**a**) MonH and (**b**) SalH.

**Figure 2 molecules-29-00364-f002:**
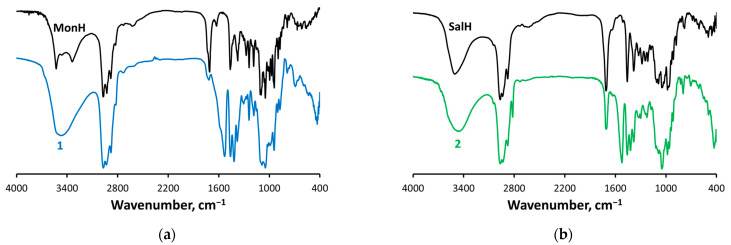
IR spectra of (**a**) MonH/complex **1** and (**b**) SalH/complex **2**.

**Figure 3 molecules-29-00364-f003:**
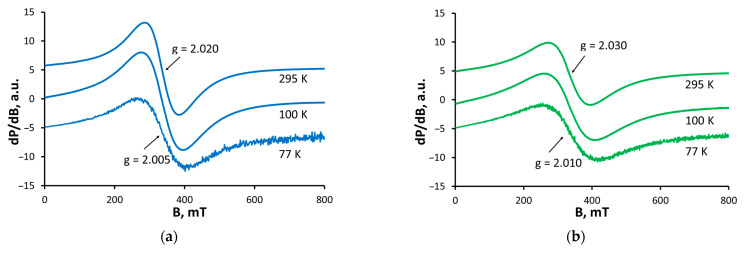
X-band temperature-dependent EPR spectra of (**a**) **1** and (**b**) **2** and the temperature dependence of the linewidth: (**c**) for **1**—150 mT (77 K), 99 mT (295 K); (**d**) for **2**—175 mT (77 K), 122 mT (295 K).

**Figure 4 molecules-29-00364-f004:**
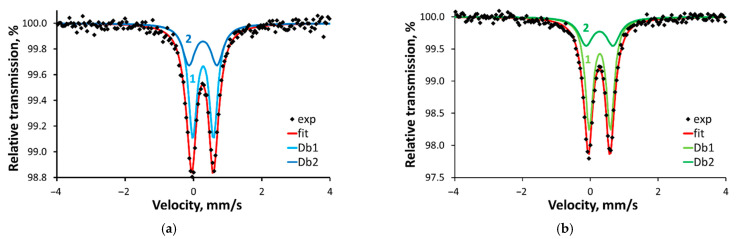
^57^Fe Mössbauer spectra of (**a**) **1** and (**b**) **2** at 293 K.

**Figure 5 molecules-29-00364-f005:**
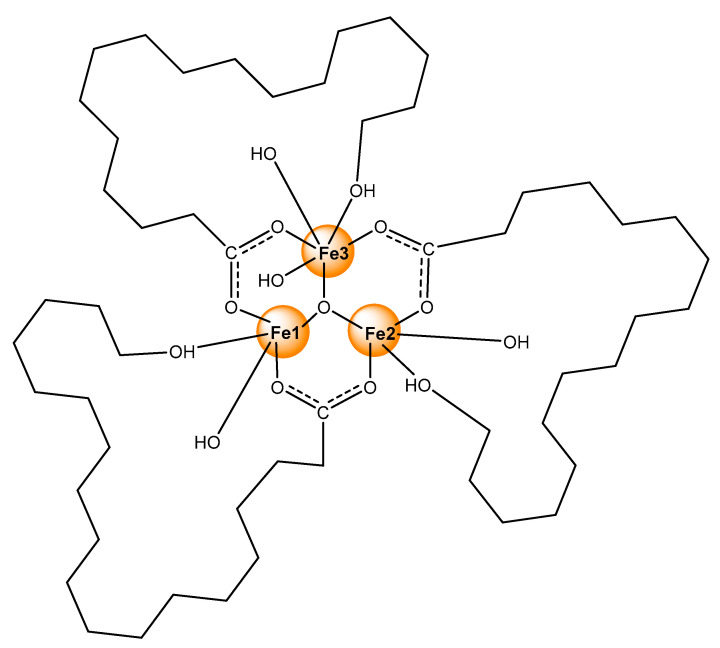
Sketch of the proposed structure of complexes **1**–**2** (note that the antibiotic backbones lie outside the plane of the Fe_3_O-unit and significant geometry distortions at each iron centre might be expected).

**Figure 6 molecules-29-00364-f006:**
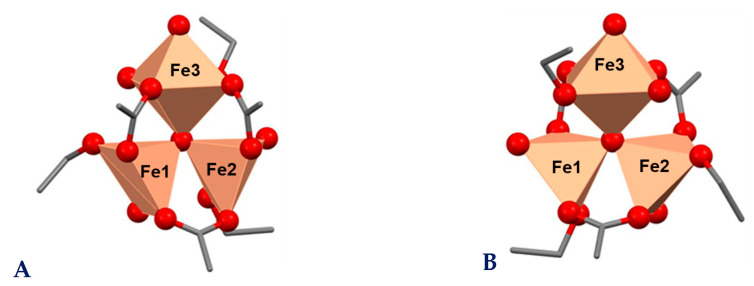
Modelled constructs of **1**–**2** with parallel (**A**) and non-parallel (**B**) orientation of the carboxylate donor atoms relative to the triangular μ_3_–O-iron plane. Hydrogen is omitted for clarity. Colour code: C—grey, O—red, Fe—orange.

**Figure 7 molecules-29-00364-f007:**
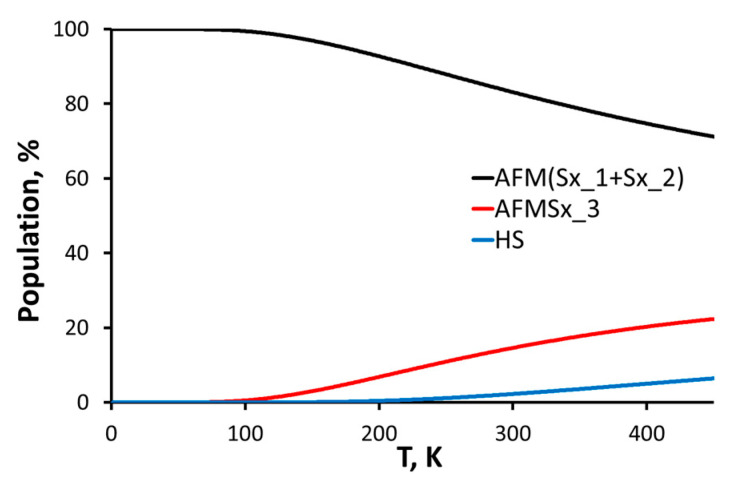
Boltzmann distribution (0−450 K) of the spin states of the construct **B**.

**Table 1 molecules-29-00364-t001:** Elemental analysis of **1**–**2.**

Complex	Composition	MW, g/mol	C, %		H, %		Fe, %	
			Calc.	Found	Calc.	Found	Calc.	Found
[Fe_3_(µ_3_-O)Mon_3_(OH)_4_], **1**	C_108_H_187_Fe_3_O_38_	2261.18	57.37	56.33	8.34	7.35	7.41	7.39
[Fe_3_(µ_3_-O)Sal_3_(OH)_4_], **2**	C_126_H_211_Fe_3_O_38_	2501.57	60.50	60.12	8.50	8.44	6.70	6.69

**Table 2 molecules-29-00364-t002:** Mössbauer spectra parameters of **1**–**2** at r. t. (the data at 77 K are given in parenthesis).

Compound	Component	δ, mm/s	∆, mm/s	Γ, mm/s	A, %	M, %
**1**	Db1	0.39 (0.59)	0.61 (0.59)	0.30 (0.28)	65 (66)	0.86 (5.21)
	Db2	0.39 (0.58)	0.82 (0.84)	0.45 (0.33)	35 (34)	0.31 (2.25)
**2**	Db1	0.39 (0.58)	0.63 (0.61)	0.28 (0.29)	70 (65)	1.69 (4.70)
	Db2	0.38 (0.57)	0.80 (0.93)	0.49 (0.38)	30 (35)	0.42 (1.95)

**Table 3 molecules-29-00364-t003:** Bond lengths [Å] for structure **B** (optimized HS state).

Bond	Fe(1)	Fe(2)	Fe(3)
μ_3-_O−Fe	1.9	1.9	2.0
COO^−^−Fe	2.0	2.0	2.1
	2.1	2.1	2.1
ROH−Fe	2.1	1.9	2.1
OH^−^−Fe	1.9	1.9	1.9
	−	−	2.1

**Table 4 molecules-29-00364-t004:** Relative energy, population, and g-factors for all spin states of construct **B**.

Multiplicity	ΔG, kcal/mol 293 K	Population, % 293 K	Population, % 77 K	g-Factor 0 K
HS	2.15	2.10	0.00	2.005
AFM(Sx_1 + Sx_2)	0.00	83.8	99.9	2.005
AFMSx_3	1.04	14.1	0.10	2.006

**Table 5 molecules-29-00364-t005:** NBO spin population of the [Fe_3_(μ_3_–O)]^7+^ core.

Centre/Spin State	HS	AFMSx_1	AFMSx_2	AFMSx_3
Fe(1)	4.43	–4.42	4.42	4.42
Fe(2)	4.43	4.43	–4.43	4.43
Fe(3)	4.49	4.48	4.49	–4.48
μ_3_–O	0.40	0.02	0.11	0.27

**Table 6 molecules-29-00364-t006:** NBO charges of the first coordination shell and metal ions.

	Fe(1)	Fe(2)	Fe(3)
Fe	2.18	2.19	2.24
μ_3_–O	–1.46	–1.46	–1.46
COO^−^	–0.82	–0.83	–0.82
COO^−^	–0.84	–0.84	–0.83
ROH	–0.85/–0.31 *	–0.83/–0.30 *	–0.97/–0.29 *
OH^−^	–1.17/–0.70 *	–1.18/–0.71 *	–1.19/–0.71 *
OH^−^	–	–	–1.02/–0.65 *

* The values are for the NBO summed OH-charges.

**Table 7 molecules-29-00364-t007:** Frequencies of some important vibrations (cm^−1^) and similarity factors (SF, the ratio of the experimental to the calculated value for each pair of results).

Vibration	Monensinate Complex			Salinomycinate Complex		
	Exp.	Calc.	SF	Exp.	Calc.	SF
$\nu_{C=O}^{asym}$, COO¯	1530	1602	0.96	1522	1533	0.99
$\nu_{C=O}^{sym}$, COO¯	1418	1501	0.94	1419	1436	0.99
$\Delta \;(\nu_{C=O}^{asym}-\nu_{C=O}^{sym})$	112	101	−	103	97	−

## Data Availability

Data are available from the authors upon request.

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
