# Peer review of "Experimental and DFT Study of Monensinate and Salinomycinate Complexes Containing {Fe33–O)}7+ Core"

_molecules, 2024, doi:10.3390/molecules29020364_

Round 1

Reviewer 1 Report

Comments and Suggestions for Authors

The reviewed work presents synthesis and characterization of novel Fe(III) complexes with two polyether antibiotics. In a lack of good monocrystals, the authors characterized the obtained compounds with several spectroscopic techniques (FTIR, EPR, Moessbauer) and proposed the structure of trinculear complex with Fe(III) ions bridged by carboxylate groups. The latter model was used in DFT calculations, which seems to reproduce well available exp. data.

The topic is interesting, the choice of exp. and computational methods is sound, and the paper is rather well written, therefore in my opinion it is suitable for publication in ‘Molecules’, however, some medium-minor corrections should be applied first.

Medium issues:

(1) If I understand correctly, among the important arguments for trinuclear complex are 1:1 metal-ligand ratio and the lack of water in the Fe coordination sphere. I think these both factors should be better exposed in the paper

(2) Are there any trinuclear Fe(III) complexes with analogous coordination sphere? (apart from ref. 44). If so, some references should be given.

(3) Was the biological activity of these complexes studied? There is a large interest in the differences in biological activity between pristine ligands and their complexes with (transition) metals. Even if the activity of complexes was lowered with respect to initial ligands, it would be quite important result.

Minor issues:

‘Results and Discussion’

(1) I think there should be a separate section with elemental and thermal analysis results (possibly 2.2, between ‘General’ and ‘Spectral properties’). Personally, I would further split ‘Spectral properties’ subsection into three subsections for each spectroscopic method (FTIR, EPR, Moessbauer) for improved clarity.

3.4 paragraph (page 10)

(1) The description of computational methodology is a bit too compact to my taste, instead of “B3LYP/6-31G(d)” and “BHandHLYP/TZVP/CP(PPP)”, I would prefer “… hybrid functional” and “… basis set”. In the latter case I guess (but I’m not sure) CP(PPP) basis set was used on Fe atom and TZVP one on remaining atoms.

(2) I would expect some justification of the choice of BHandHLYP functional for spin state energetics (comment and/or references).

(3) DFT optimized models should be included in SI (preferably .xyz files).

(4) More details about the free energy calculations are needed (I suppose that the contributions due to vibrational effects in harmonic approx. were included, but again I’m not sure).

Some proposals for future studies:

(1) For future structural studies on the coordination compounds in non-crystalline form it would be helpful to include X-ray absorption spectroscopy, particularly EXAFS enables detailed information about the coordination sphere of scattering atom. This method of course requires the access to synchrotron facility (or collaboration with the group having such access).

(2) The prediction of spin state energetics is a tricky business, it would be worthwhile to include calculations obtained with advanced wavefunction based methods level, like CASSCF (+PT2).

(3) p.6, l. 206 „The total structures of complexes 1-2 are oversized for first principles quantum chemical calculations.” This statement is principally true for the optimization at hybrid functional level (though probably even then the optimization is not impossible with current computational resources), however, such large complexes could probably be optimized in moderate size basis set at DFT level using gradient functionals.

Reviewer 2 Report

Comments and Suggestions for Authors

1.The quality of all figs should be improved, especially Fig 3, Fig 7.

2. The authors have stated “Figure 5. Sketch of the proposed structure of complexes 1-2 .” This means the two complex have the similar structure? Why?

3. Pls also give the main peak on the figure from IR.

4. Some work on the DFT could be compared and discussed, such as Theor. Chem. Acc. 2022, 141, 68. Monatsh. Chem, 2019, 150, 1355–1364 and Monatsh Chem., 2017, 148, 1269-1276.

5. There are several spelling errors, grammar issues and wrong syntaxes.

6. Pls also give the TGA and EA for checking the final formula for the two complexes.

7. Pls also provide the XPS for checking its oxide state for Fe.

Comments on the Quality of English Language

check

Round 2

Reviewer 2 Report

Comments and Suggestions for Authors

accept